# A comparative study between Near-Infrared (NIR) spectrometer and High-Performance Liquid Chromatography (HPLC) on the sensitivity and specificity

**Elisa M. Maffioli**[1]*, **Chimezie Anyakora**[2]

**1** Department of Health Management and Policy, University of Michigan, School of Public Health, Washington Heights, Ann Arbor, Michigan, United States of America, **2** Bloom Public Health, No 4, Thabo Mbeki Close, Off TY Danjuma Street, Asokoro, Abuja, Nigeria and School of Science and Technology, Pan Atlantic University, Lagos, Nigeria

* elisamaf@umich.edu

## Abstract

It is estimated that 10.5% of medicines in low- and middle-income countries are substandard or falsified (SF), causing approximately 1 million deaths annually. Over the past two decades, there have been significant technological advancements in low-cost, portable screening devices to detect poor-quality medicines, which could be especially beneficial in these countries. The pharmaceutical market in Nigeria is valued at USD 4.5 billion and is growing at over 9% annually. However, SF medicines remain a major public health concern. We compared a novel Near-Infrared (NIR) Spectrometer with high-performance liquid chromatography (HPLC) by analyzing 246 drug samples purchased from retail pharmacies across the six geopolitical regions of Nigeria. We measured the sensitivity and specificity of a patented and Artificial Intelligence (AI) - powered handheld NIR spectrometer, which uses a proprietary machine-learning algorithm as well as hardware and software, across four categories of medicines: analgesics, antimalarials, antibiotics, and antihypertensives. Our findings reveal that the prevalence of SF medicines remains high, with 25% of samples failing the HPLC test. When tested with the NIR spectrometer, only a smaller subset of medicines—specifically analgesics—failed the test. Sensitivity and specificity for all medicines were 11% and 74%, respectively. For analgesics, the sensitivity was 37%, and the specificity was 47%. While these devices hold great potential, regulators should require more independent evaluations of various drug formulations before implementing them in real-world settings. Improving the sensitivity of these devices should be prioritized to ensure that no SF medicines reach patients.

## Background

Substandard and falsified (SF) drugs remain a major public health concern. It is estimated that between 10% and 14% of medicines in low- and middle-income countries (LMICs) are SF, leading to an estimated 1 million deaths annually [1, 2]. The prevalence of SF medicines

**Data availability statement:** The datasets used and/or analyzed during the current study are in Supplementary Information.

**Funding:** The project was funded by USAID-DIV (7200AA21FA00006) and internal funding from the University of Michigan. The funders had no role in study design, data collection and analysis, decision to publish, or preparation of the manuscript.

**Competing interests:** The authors have declared that no competing interests exist.

**Abbreviations:** AI, Artificial Intelligence; API, Active Pharmaceutical Ingredients; HPLC, High-Performance Liquid Chromatography; LMICs, Low- and Middle-Income Countries; NAFDAC, National Agency for Food and Drug Administration and Control; NIR, Near-Infrared against; SF, Substandard and Falsified

varies by country and drug type, with Sub-Saharan Africa and Southeast Asia experiencing the highest rates, especially for antimalarials and antibiotics [1,3,4].

SF medicines pose significant risks to patients, including adverse effects from incorrect active pharmaceutical ingredients (API), treatment failure, prolonged illness, and preventable deaths. They also contribute to antimicrobial resistance [5]. From a health systems perspective, SF medicines increase care costs, place added strain on providers and erode trust in the health system. A higher disease burden can lead to income loss for patients, reduced productivity, and greater poverty.

SF medicines are difficult to detect. The illegal trafficking of poor-quality medicines remains a highly profitable business—valued between $200 and $431 billion [6]. This issue is especially prevalent in LMICs, where high prices limit access to authentic medicines, and inadequate government oversight allows illegal suppliers to go unpunished [7–9]. Over the past two decades, several low-cost, portable devices have been developed to help regulatory authorities detect poor-quality medicines [10]. However, their accuracy remains limited.

Our study was conducted in Nigeria, where the pharmaceutical market is valued at USD 4.5 billion and is growing at 9% annually [11]. The country is highly import-dependent, sourcing 70% of its finished products from abroad, and it relies almost entirely on other countries for API for local manufacturing [12]. Analgesics account for the largest market share (25%), followed by antibiotics (15%), multivitamins (15%), antimalarials (14%), and antihypertensives (8%) [13].

Several methods have been developed to determine the authenticity of SF medicines. Traditional laboratory analysis is costly, labor-intensive, and requires sample transport, preparation, and expert handling. In recent decades, portable testing devices and mobile mini labs have allowed for testing in remote areas, though many are still expensive, complex, and lack real-time data capabilities. Handheld spectrometers offer promise but are often costly and heavy. Thus, lower-cost, portable screening tools are being developed for use by regulators to detect poor-quality drugs at key points in the supply chain [14]. While the Nigerian National Agency for Food and Drug Administration and Control (NAFDAC) has gained international recognition in the use of cutting-edge technologies such as Raman Spectroscopy, GPHF Minilab and Mobile Authentication Service, the number of deployable technologies is insufficient for such a large population. In the fight against SF medicines, NAFDAC just launched a "Green Book" to verify drugs in January 2024, i.e., "a database of 6432 registered and approved drugs for sale and distribution" [15].

We present insights from a comparative study between a proprietary Near-Infrared (NIR) Spectrometer and high-performance liquid chromatography (HPLC). To protect proprietary information and sensitive business data that could potentially affect the company's competitive position, the company's identity has been anonymized in this study.

## Methods

We tested the validity of this NIR spectrometer against HPLC. The device is a patented and Artificial Intelligence (AI) - powered handheld spectrometer, which uses a proprietary machine-learning algorithm as well as hardware and software with a NIR-Dispersive range of 750 to 1500nm.

The spectrometer analyzes a drug's spectral signature to detect poor-quality medicines by comparing it to a cloud-based AI reference library of spectral signatures. The NIR is capable of detecting both substandard and counterfeit drugs. The device captures the spectrum of the entire drug (API and excipients) and stores the spectral signature of this medical product. It also captures the intensity of the spectrum, which is proportional to that of the authentic product from the manufacturer. The device can determine counterfeit drug samples by

matching the signature spectrum of the reference product with the drug sample collected in the field. If the spectrum of the pill collected in the field differs from the spectrum of the authentic sample, the spectrometer displays a "non-match" result. The device is also able to detect substandard drug samples by matching the intensity of the reference product with the drug sample collected in the field. Researchers were not given details on other parameters or thresholds, which may vary by drug. The process takes about 20 seconds, with a quality report sent to a smartphone app.

Customized chemometric models are necessary to develop the reference library, with authentic samples sourced for a fee by the company. The company claimed at the time that any product (in pill form) could be analyzed with their NIR device, provided the reference library was updated. For this specific study, the company was responsible for sourcing the exact branded drug samples and dosage forms (though not the lot numbers tested in the field). More specifically, the company stated that 3 out of the 20 drugs in our sample were already included in their library: May & Baker Para (Paracetamol) Tabs. (x96), Emzor Paracetamol Tabs., 500mg, and Lonart-DS Artemether Lumefantrine Tabs., 80mg/480mg. The remaining drug samples were sourced and added to the reference library by the company. The company did not share the results of their training exercises. Unlike HPLC, this portable device does not require sample destruction and enables real-time analysis, making it accessible for regulators, law enforcement, customs, manufacturers, and pharmacies.

In November 2022, we purchased medicine samples from randomly selected pharmacies in rural and urban areas of the six largest cities in Nigeria's geopolitical zones: Abuja, Kano, Lagos, Onitsha, Port Harcourt, and Yola. Twelve enumerators, acting as mystery shoppers, began at recorded locations and conducted random walks to locate pharmacies and were instructed to purchase a randomly selected branded drug from a list of 20. All drugs were tested using the NIR spectrometer, and a sub-sample (N = 246) was selected as a weighted average by drug category, reflecting the proportions found by mystery shoppers in pharmacies, and sent to a laboratory for HPLC compositional quality analysis between December 2022 and February 2023 (Table 1). This sub-sample excluded multivitamins, which were less common in the pharmacies sampled (Table 2).

We conducted HPLC analysis at Hydrochrom Analytical Services Limited, located in Gowon Estate, Lagos. Drug samples were initially collected at our partner's office in Abuja, categorized, and then shipped to the laboratory. The HPLC analysis was performed on an Agilent 1100 HPLC system equipped with an online degasser, variable UV detector, quaternary pump, autoliquid sampler, and a thermostated column compartment. Chromatographic data were processed using Chemstation Rev. B.04.03-SP1 software. A validated method was employed for each molecule, depending on its specific requirements. Prior to each analysis, system suitability was confirmed using a reference standard for each analyte. S1 Table presents the analytical parameters for linearity, correlation and detection limits of compounds by HPLC. S2 Table describes the sample preparation and analytical conditions for each analyte.

We compared the results from the NIR spectrometer and HPLC to measure sensitivity and specificity by medicine category. Sensitivity is the proportion of medicines detected as poor quality by the NIR spectrometer out of all those identified as poor quality by HPLC (true positives/[true positives+false negatives]). Specificity is defined as the proportion of medicines identified as authentic by the NIR spectrometer out of all those determined to be good quality by HPLC (true negatives/[true negatives+false positives] [16]. Additionally, we calculated the positive predictive value (true positives/ [true positives + false positives]) and negative predictive value (true negatives/ [true negatives + false negatives]).

**Table 1. Description of medicines purchased.**

| | Dosage | Ingredients | N samples tested |
|---|---|---|---|
| **Analgesics** | | | |
| Panadol Extra | 500mg/65mg | Paracetamol/Caffeine | 28 |
| May & Baker Paracetamol | 500mg | Paracetamol | 17 |
| Emzor Paracetamol | 500mg | Paracetamol | 51 |
| Tuyil Cenpain Night | 500mg/25mg | Paracetamol/Diphenhydramine HCL | 14 |
| **Antibiotics** | | | |
| Swipha Tiniflox Tinidazole + Ofloxacine | 600/200mg | Tinidazole/Ofloxacin | 4 |
| Sanofi Avensis Flagyl Metronidazole | 400mg | Metronidazole | 7 |
| Fidson Ciprotab (Ciprofloxacin) | 500mg | Ciprofloxacin | 14 |
| Neimeth Pyrantrin Pyrantel Pamoate | 125mg | Pyrantel Pamoate | 13 |
| **Antihypertensives** | | | |
| Normoretic | 5mg/ 50mg | Amiloride HCL/Hydrochlorothiazide | 12 |
| Bonduretic | 5mg/ 50mg | Amiloride HCL/Hydrochlorothiazide | **9** |
| Nifedin Nifedipine Dexcel | 20mg | Nifedipine | 8 |
| Bondomet Methyldopa | 250mg | Methyldopa | **2** |
| **Antimalarials** | | | |
| Swipha Swidar Sulphadoxine + Pyrimethamine | 500/25mg | Sulphadoxine/Pyrimethamine | 16 |
| Lonart-DS Artemether Lumefantrine | 80mg/480mg | Artemether/Lumefantrine | 23 |
| Coartem Artemether Lumefantrine | 80mg/480mg | Artemether/Lumefantrine | 14 |
| Artequick Artemisinin Piperaquine | 62.5mg/375mg | Artemisinin/Piperaquine | 14 |
| **Multivitamins** | | | |
| Kunimed Ascomed Vitamin Coloured Vitamin C | 100mg | Vitamin C | N/A |
| Vitamin C (Chemo-Pharma) White | 100mg | Vitamin C | N/A |
| Chemo-Pharma Vitamin C Colored | 100mg | Vitamin C | N/A |
| sMeyer B Complex | 100mg | Vitamin B | N/A |

**Table 2. Sample of medicines tested, by category.**

| Categories | % | N |
|---|---|---|
| Analgesics | 44.72 | 110 |
| Antibiotics | 15.45 | 38 |
| Antihypertensives | 12.60 | 31 |
| Antimalarials | 27.24 | 67 |
| **Total** | **100** | **246** |

## Ethics Approval

This study was approved by the University of Michigan (HUM00214684) and the National Health Research Ethics Committee (NHREC) in Nigeria (NHREC/01/01/2007). Oral or written consent was not required as this study does not involve human subjects and falls outside the scope of human subjects' research.

## Results

Enumerators visited 1,296 pharmacies and successfully purchased one drug from the random list in 93.7% (N = 1,214) of cases. All medicines were tested with the NIR spectrometer (N = 1,142), and a sub-sample (N = 246) was also tested with HPLC. The NIR spectrometer

identified 4.8% (N = 55) of the samples as failing the test, all of which were analgesics. Among the 246 tested, 22% (N = 55) failed the NIR spectrometer (S3 Table). When tested with HPLC, 25% (N = 62) failed due to API falling outside the 90–110% range, indicating a high prevalence of poor-quality medicines. Of those failing the HPLC test, 35% (N = 22) were antihypertensives, 31% (N = 19) were analgesics, 19% (N = 12) were antibiotics, and 15% (N = 9) were antimalarials [17].

Table 3 compares the passing and failing rates between the two tests. The overall sensitivity of the NIR spectrometer was 11%, with a specificity of 74%. The positive predictive value was 13%, and the negative predictive value was 71%. These metrics varied by drug type (S4 Table): for analgesics, sensitivity was 37% and specificity was 47%, while the NIR spectrometer performed poorly for other drug categories, with sensitivity at 0% and specificity at 100%. A highly sensitive test effectively detects SF medicines by minimizing false negatives, ensuring that SF medicines are not missed and mistakenly classified as genuine. Conversely, a highly specific test reduces false positives, accurately identifying genuine medicines without incorrectly labeling them as SF. Although low specificity can lead to additional costs and work in reference laboratory assays, prioritizing higher sensitivity is essential to prevent SF medicines from reaching patients.

## Discussion

In this study, we build on existing evidence that reviews field detection devices for screening medicine quality, including evaluations of both laboratory and field devices. Previous research has underscored the potential and limitations of different devices in detecting SF medicines under controlled and field conditions [14,18–22]. Our study contributes to this body of work by testing a novel, low-cost technology designed to offer real-time data, which could represent a significant advancement if proven viable. This technology, if effective, holds promise for reducing dependency on traditional laboratory testing by allowing immediate screening of pharmaceutical products in the field, potentially strengthening quality assurance processes in remote or resource-limited settings. However, our findings indicate that this technology is not yet suitable for real-world deployment.

Our comparative exercise, combined with expert discussions, highlighted key considerations for further investment in these technologies and their use by regulatory authorities. First, the low sensitivity of the NIR spectrometer in our study may align with findings from laboratory evaluations of 12 portable devices, which showed high sensitivities for detecting medicines with no or incorrect APIs but variable sensitivities (0% to 100%) for samples with

**Table 3. Sensitivity and Specificity of NIR spectrometer.**

| | | HLPC Lab | | |
| --- | --- | --- | --- | --- |
| | | **Fail** | **Pass** | |
| **NIR spectrometer** | Fail | True positives (TP) | False positives (FP) | |
| | | N = 7 | N = 48 | N = 55 |
| | Pass | False negatives (FN) | True negatives (TN) | |
| | | N = 55 | N = 136 | N = 191 |
| | Total | N = 62 | N = 184 | N = 246 |
| Sensitivity | TP/(TP+FN) | | | 11% |
| Specificity | TN/(TN+FP) | | | 74% |

*Notes: N* represents the number of drug samples in each cell.

50% to 80% API [18]. Our results suggest the NIR spectrometer may require further technological improvements, especially for detecting substandard formulations.

Second, adjusting the pass/fail threshold for the test may be necessary. Increasing the threshold could improve sensitivity but reduce specificity [20]. Public health concerns suggest that enhancing sensitivity for detecting SF medicines is crucial, even at the expense of specificity. This is particularly important when considering the potential harm that SF medicines can cause to patients, including adverse reactions or treatment failures. Therefore, policymakers and regulatory bodies must carefully weigh the trade-off between sensitivity and specificity, taking into account the broader implications for patient safety and public trust in the healthcare system.

Third, building and maintaining an up to date "spectral reference library" with quality-assured genuine samples is both challenging and costly. Authentic materials are often difficult to obtain and may not be suitable for screening finished products, as spectra can be influenced by both APIs and excipients and can vary between brands [14]. In our study, the company was responsible for building a customized reference library using authentic samples procured directly from the manufactures. The researchers did not have control over this process, and the company did not share the results of their procedure. As a result, we cannot be certain that the agreed-upon process was followed, although we have no reason to believe otherwise. Our findings suggested that the NIR spectrometer's AI algorithm was better trained on analgesics, highlighting the need for a comprehensive and shareable "spectral reference library" to facilitate effective device training and comparison.

Our study is not without limitations. First, our comparative study is limited to 246 drug samples purchased in Nigeria, which may not generalize to other branded or non-branded medicines or to different geographical contexts. Second, due to the proprietary nature of the machine-learning algorithm and the hardware and software, and to protect proprietary information and sensitive business data, we are unable to disclose the company's identity. This constraint limits the researchers' ability to fully share technical details and methodologies related to the device's design, calibration, and performance metrics. Future research should seek to test this and similar devices across a broader range of medicines, regions, and conditions to validate their efficacy and robustness in diverse real-world settings.

## Conclusion

This study addresses the critical issue of ensuring the reliability of devices, such as Near-Infrared (NIR) spectrometers, for screening substandard and falsified (SF) medicines. We compared a novel NIR spectrometer to high-performance liquid chromatography (HPLC) by analyzing 246 drug samples purchased from retail pharmacies across the six geopolitical regions of Nigeria. We measured the device's specificity and sensitivity.

Our findings indicate that this device is not yet ready for field deployment in the context studied, as it demonstrated particularly low sensitivity, limiting its ability to prevent SF medicines from reaching patients. The results align with existing evidence, where no current device has demonstrated the capability to assess the quality of active pharmaceutical ingredients (APIs) across the diverse range of formulations and conditions encountered in real-world settings [14]. This comparative study highlights significant gaps in the regulatory framework and a lack of standardization for evaluating such devices, as observed in this specific setting.

First, we noted limited transparency from the manufacturer regarding data analytics methods and device validation processes. We recommend that manufacturers provide detailed information on data processing methods and validation procedures to enable independent evaluation of device reliability. Additionally, regulatory authorities should establish clear,

standardized performance criteria for screening devices, including field-testing protocols tailored to the study region.

Second, the study revealed that manufacturers rely on limited laboratory datasets without sufficient field testing. We suggest that devices with low sensitivity or inadequate validation undergo additional laboratory and field evaluations in similar settings before deployment to ensure they can effectively screen SF medicines. Collaboration among regulators, researchers, and manufacturers may support the development and refinement of screening technologies suited to specific contexts.

By addressing these gaps and implementing these interventions, stakeholders can improve the reliability of NIR spectrometers, thereby enhancing patient safety and the integrity of pharmaceutical supply chains. Our findings underscore the importance of further research and rigorous validation to ensure these devices meet their intended purpose.

## Supporting Information

**S1 Table. Analytical Parameters for Linearity, Correlation, and Detection Limits of Compounds by High-Performance Liquid Chromatography.**
(DOCX)

**S2 Table. Sample Preparation and HPLC Parameters for Compound Analysis.**
(DOCX)

**S3 Table. Passing and failing rates by test and category of medicines.**
(DOCX)

**S4 Table. Sensitivity and Specificity of NIR spectrometer, by category of medicines.**
(DOCX)

## Acknowledgements

We thank the team at Bloom Public Health who collected the drug samples and gathered the data.

## Author contributions

**Conceptualization:** Elisa M Maffioli, Chimezie Anyakora.

**Data curation:** Elisa M Maffioli.

**Formal analysis:** Elisa M Maffioli.

**Funding acquisition:** Elisa M Maffioli.

**Methodology:** Elisa M Maffioli, Chimezie Anyakora.

**Project administration:** Chimezie Anyakora.

**Supervision:** Elisa M Maffioli, Chimezie Anyakora.

**Validation:** Elisa M Maffioli, Chimezie Anyakora.

**Writing – original draft:** Elisa M Maffioli.

**Writing – review & editing:** Chimezie Anyakora.

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
