## [Decision Letter · Decision Letter 0]

22 Dec 2024

PONE-D-24-54221Validation of Near-Infrared Spectrometer to detect Substandard and Falsified MedicinesPLOS ONE

Dear Dr. Maffioli,

Thank you for submitting your manuscript to PLOS ONE. After careful consideration, we feel that it has merit but does not fully meet PLOS ONE’s publication criteria as it currently stands. Therefore, we invite you  to carefully and objectively improve your manuscript following all the reviewer comments and submit a revised version of the manuscript that addresses the points raised during the review process.

We look forward to receiving your revised manuscript.

Kind regards,

Hope Onohuean, PhD

Academic Editor

PLOS ONE

Journal Requirements:

2. Thank you for stating the following financial disclosure: The project was funded by USAID-DIV (7200AA21FA00006) and internal funding from the University of Michigan.

3. In the online submission form, you indicated that the datasets used and/or analyzed during the current study are available from the corresponding author on reasonable request.

Reviewers' comments:

Reviewer's Responses to Questions

**Comments to the Author**

1. Is the manuscript technically sound, and do the data support the conclusions?

Reviewer #1: No

Reviewer #2: Partly

Reviewer #3: No

2. Has the statistical analysis been performed appropriately and rigorously? 

Reviewer #1: I Don't Know

Reviewer #2: Yes

Reviewer #3: I Don't Know

3. Have the authors made all data underlying the findings in their manuscript fully available?

Reviewer #1: No

Reviewer #2: No

Reviewer #3: No

4. Is the manuscript presented in an intelligible fashion and written in standard English?

Reviewer #1: Yes

Reviewer #2: Yes

Reviewer #3: Yes

5. Review Comments to the Author

Reviewer #1: The paper submitted by Maffioli and Anyakora discusses the « validation » of a NIR spectrometer to detect SF medicines.

As mentioned by the authors, rigorous and transparent validation of screening devices such as NIR handheld spectrometers is mandatory before using it in routine.

However, the paper is anything but transparent. No information is given on the device used, the algorithms used, how the models have been trained and no detail is given on the obtained results. Therefore, it is impossible to understand what the limitations of this technique are and what may be improved. This paper, in its present form, cannot be published since it does not add any new information and may be prejudicial to the reputation of promising techniques such as NIR spectroscopy. Indeed, each technology comes with its own advantages and drawbacks. This is why is it imperative to explain what the objective of the models was and what the results are.

Detailed comments:

- L111: what do the authors mean when they say that handheld devices are not field ready? The objective of these devices is to be used in the field, and they are efficiently used in many field settings since a long time.

- L112: what do the authors mean by “low-cost”? This appears subjective.

- L121: what is a “validation study”? The present article does not follow any validation guidelines. Please refer to the existing guidelines for validation or do not use this terminology.

- The anonymization of the whole NIR part is not acceptable since it is the core of the paper. It is impossible to the reviewer to understand the few information provided. Where the measures performed in reflectance or transmittance? What is the kind of detector, how is performed the white reference, … It is mentioned that the device is “similar to NIR-S-G1” but the spectral range is completely different… In which sense is it similar? Regarding the algorithms used, just calling it “AI” is insulting for the reader. What is this? Is it a regression approach to quantify the API? Is it a qualitative approach to authenticate a brand or to identify an API? How are the algorithms trained? How have they been validated? This is really the core of the device and no information at all is provided.

- There is no detail on the dataset. Only 256 spectra are mentioned in the tables but at L186, 1142 samples have been tested with NIR. Where are the remaining 886 spectra ?

- The dataset is briefly described but once again no details. How many samples are collected per brand? Are there different batches?

- The HPLC results should also be provided. 62 samples where outside the 90-110% range. But how far? 89% or 25% is a totally different story. NIR in reflectance on finished product is not a magic tool. The results will depend on how the system is calibrated. Once again, we have no information.

- No information is provided on the fact that the API was correctly present or not. Where all samples containing the correct API?

- The detection of substandards imply the quantitation of the samples. How was this performed? NIR only “sees” API/excipients ratios. Therefore, have the masses of the tablets been measured?

Reviewer #2: The authors compared the quality of 246 medicines using an inexpensive handheld spectrophotometer and a gold standard method, HPLC. The medicines were collected in Nigeria. While the finding that about 25% of the products failed HPLC assay is not unexpected given the wealth of previous research about poor quality medicines in LMICs including Nigeria, this manuscript highlights the pitfalls of the currently unregulated market for devices like the NIR spectrophotometer, that claim to detect substandard or falsified pharmaceuticals (SFPs). The prior literature is treated fairly and the writing is concise and well organized. I do have some concerns about the manuscript, described below, but I think the paper provides very useful information that could save other researchers and regulators a lot of trouble in future efforts to detect SFPs. I hope the authors revise and resubmit.

The methodology is described clearly, with one exception: it's not clear what discussions the authors had with the manufacturer about the product's training/library, or what the unnamed manufacturer claimed about their product's capabilities.

A1. The method section says that the company "produces a customized reference library with authentic samples sourced for a fee".

describe what samples went into the reference library that was used in this study. To be specific, was the library trained with the same exact brands/lot numbers as the samples that were tested in the field? Did the company do any testing to make sure their training samples were actually good quality? That's been a problem in previous studies. If not the same brands, were they the same dosage forms (eg, tablets with same dose of API)? For samples like the analgesics where several brands contained multiple drugs, were correct reference samples used in the NIR library?

If it's not known what samples were in the reference library (eg, because the company will not disclose) that needs to be stated, along with any claims the company makes about what products can be analyzed with their NIR.

A2. In the Methods section, the authors state that the non-match result from the NIR is a result of "alternative excipients, APIs, or contaminants". However, the HPLC results do not show that any of the products contained alternative excipients, APIs, or contaminants. The HPLC results are just assays.

 Do the manufacturers actually make the claim that the NIR spectrophotometer can detect substandard products? If not, there needs to be some discussion of why the authors are calling it a failure of the instrument when it missed substandard drugs. I think they could make this argument even if the mfr does not make the claim, as it is a serious problem if the NIR cannot detect a product with very low API content, eg <50%, but there should be some acknowledgement that "s" and "f" products are different challenges.

B. The HPLC results are presented only in aggregate (eg, 50% of the analgesics failed to meet the HPLC content standard, but it's not clear which products failed or how badly they failed).

Please provide data on the individual products in the supporting information section. It would be ideal to list the full product metadata (brand, lot number, expiration date, API content found, NIR result). A standard circumlocution which protects the researcher is to say that the product is "stated to be Brand Wonderful, lot number 12345"--in this case, you are just reporting the information that is printed on the packet or blister pack. If the brand and lot number metadata cannot be disclosed, that should be explained (eg, "due to pending regulatory activity we were asked not to disclose brand names"). In this case, the brands could be coded as Brand A, B, C, etc, and expiration dates, API contents, and NIR results could be provided.

C. It is not stated whether the chemical analysis results for failing products (the HPLC, not the NIR) were reported to the applicable regulatory agency, which seems to me to be ethically necessary. This could be mentioned in the "methods" section

please report whether data about products that failed HPLC assay was shared with NAFDAC and/or WHO.

There are some minor comments for the authors:

D1. In the introduction, the term "counterfeit" is thrown around a lot. However, counterfeit does not mean "bad quality", any more than "authentic" means good quality--there are counterfeit versions of pharmaceuticals that are nearly identical to the original product, and there are a lot of products that were produced by their stated manufacturer that have dreadful quality, eg substandard amount of API, the wrong API, or wrong excipients. Since neither HPLC nor NIR claim to detect counterfeits anyway, can you just take it out?

Please rewrite that third paragraph of the intro, using "substandard and falsified" or "bad quality".

D2. The "Food and Drug Administration 1994" reference doesn't seem like the greatest reference for the value of NIR since it is 30 years old--instruments and data analytics have advanced a lot since the 90's!

find a more current reference for the value of NIR in pharma screening

D3. table A3--header says HPCL, should be HPLC.

Reviewer #3: Title and content: “Validation of Near-Infrared Spectrometer to detect Substandard and Falsified Medicines”. This is confusing, and it doesn’t match the content. The title should be defined in the way it matches the content and defines the goal and scope of the study. Line 173-180 show that you have compared results by two different analytical techniques/equipments. In my understanding, you should target to validate an analytical method, and consider the method validation parameters as defined by ICH. In case you were interested in comparing the two analytical techniques/equipments, you should formulate the title appropriately and avoid the word “validation”. Therefore, define a title that matches the content of your work and findings. Even if we assume that the NIR apparatus corresponds to an analytical method, you shouldn’t claim to have validated it while you have considered only two parameters! You have done a good work, but it needs an appropriate title. Another issue is the comparison of a qualitative analytical technique (NIR-AI) to a quantitative analytical technique (HPLC): the NIR-AI is designed for the screening step, to detect deviations from the specific original/genuine formulations; but, HPLC quantifies the content of an specific active pharmaceutical ingredients using specific validated analytical methods. Suggested title: A comparative study between NIR and HPLC on the specificity and selectivity.

Line 93-95: “This issue is especially prevalent in LMICs, where high prices limit access to authentic medicines, and inadequate government oversight allows counterfeiters to go unpunished”. This statement requires at least one supporting reference, you can’t just say this.

Line 139-142: “Unlike HPLC, this portable device does not require sample destruction and enables real-time analysis, making it accessible for regulators, law enforcement, customs, manufacturers, and pharmacies (Food and Drug Administration, 1994)”. This reference doesn’t support this statement. You should also avoid references older than 10 years.

Line 191-193: “Of those failing the HPLC test, 35% (N=22) were antihypertensives, 31% (N=19) were analgesics, 19% (N=12) were antibiotics, and 15% (N=9) were antimalarials (Maffioli et al. 2024)”. Unless this manuscript has been already published eleswhwre, otherwise this reference should not be used to support your findings.

Conclusion (line 265-281): it is too much general. Kindly formulate a conclusion bound to your data/findings, keeping in mind the scope of your study. End the conclusion pointing out the identified gaps and suggesting interventions to address/fill in the gaps.

6. PLOS authors have the option to publish the peer review history of their article (what does this mean? ). If published, this will include your full peer review and any attached files.

**Do you want your identity to be public for this peer review?** For information about this choice, including consent withdrawal, please see our Privacy Policy .

Reviewer #1: No

Reviewer #2: **Yes: ** Marya Lieberman

Reviewer #3: **Yes: ** Thomas Bizimana

---

## [Decision Letter · Decision Letter 1]

28 Jan 2025

PONE-D-24-54221R1A Comparative Study Between Near-Infrared (NIR) Spectrometer and High-Performance Liquid Chromatography (HPLC) on the Sensitivity and Specificity.PLOS ONE

Dear Dr. Maffioli,

Thank you for submitting your manuscript to PLOS ONE. After careful consideration, we feel that it has merit but does not fully meet PLOS ONE’s publication criteria as it currently stands. Therefore, we invite you to submit a revised version of the manuscript that addresses the points raised during the review process.

We look forward to receiving your revised manuscript.

Kind regards,

Hope Onohuean, PhD

Academic Editor

PLOS ONE

Journal Requirements:

**Additional Editor Comments:**

One of the reviewer has some comments, which i will like you to revised, as mentioned below.

General: Make the lines numbers continuous to ease reference for reviewers’ comments. For example, some lines under the “Conclusion” section are not numbered, same for references.

Line 132-133: “Manufactured in Taiwan, it was priced under $3,000 - significantly cheaper than existing spectrometry devices- or available for rent”. This looks like an advertisement for this machine/brand, and it would not be scientifically good. I recommend to avoid such statements, especially the mention of the price.

Line 208-210: “Of those failing the HPLC test, 35% (N=22) were antihypertensives, 31% (N=19) were analgesics, 19% (N=12) were antibiotics, and 15% (N=9) were antimalarials (Maffioli et al. 2024)”. My previous comment was not addressed. It seems like you are making reference you this manuscript, while not yet (I assume) published!

Line 285-295: My previous comment was not addressed. The conclusion is still too much general, not reflecting the outcome of your study. You should learn how to formulate a good conlusion and apply the principles here. Also refer to my previous comment and improve this important part of your manuscript. Please note that most of readers do not have time to read tables and figures, they go strait to the conclusion and should find this part helpful in terms of answers to your research questions of your hypothesis. What were the study problem? What were the identified gaps? What are the suggested interventions to address these gaps? This section should also be concise and specific to your study findings. Avoid generalizing or extrapolating the findings to other settings that were not part of your study.. 

Reviewers' comments:

Reviewer's Responses to Questions

**Comments to the Author**

1. If the authors have adequately addressed your comments raised in a previous round of review and you feel that this manuscript is now acceptable for publication, you may indicate that here to bypass the “Comments to the Author” section, enter your conflict of interest statement in the “Confidential to Editor” section, and submit your "Accept" recommendation.

Reviewer #2: All comments have been addressed

Reviewer #3: (No Response)

2. Is the manuscript technically sound, and do the data support the conclusions?

Reviewer #2: Yes

Reviewer #3: Yes

3. Has the statistical analysis been performed appropriately and rigorously? 

Reviewer #2: Yes

Reviewer #3: I Don't Know

4. Have the authors made all data underlying the findings in their manuscript fully available?

Reviewer #2: Yes

Reviewer #3: No

5. Is the manuscript presented in an intelligible fashion and written in standard English?

Reviewer #2: Yes

Reviewer #3: Yes

6. Review Comments to the Author

Reviewer #2: The changes made to the mss give a much better picture of the manufacturer's claims, and the supporting information now provides the raw data supporting the results of this comparative study.

Reviewer #3: Re-review comments

General: Make the lines numbers continuous to ease reference for reviewers’ comments. For example, some lines under the “Conclusion” section are not numbered, same for references.

Line 132-133: “Manufactured in Taiwan, it was priced under $3,000 - significantly cheaper than existing spectrometry devices- or available for rent”. This looks like an advertisement for this machine/brand, and it would not be scientifically good. I recommend to avoid such statements, especially the mention of the price.

Line 208-210: “Of those failing the HPLC test, 35% (N=22) were antihypertensives, 31% (N=19) were analgesics, 19% (N=12) were antibiotics, and 15% (N=9) were antimalarials (Maffioli et al. 2024)”. My previous comment was not addressed. It seems like you are making reference you this manuscript, while not yet (I assume) published!

Line 285-295: My previous comment was not addressed. The conclusion is still too much general, not reflecting the outcome of your study. You should learn how to formulate a good conlusion and apply the principles here. Also refer to my previous comment and improve this important part of your manuscript. Please note that most of readers do not have time to read tables and figures, they go strait to the conclusion and should find this part helpful in terms of answers to your research questions of your hypothesis. What were the study problem? What were the identified gaps? What are the suggested interventions to address these gaps? This section should also be concise and specific to your study findings. Avoid generalizing or extrapolating the findings to other settings that were not part of your study.

7. PLOS authors have the option to publish the peer review history of their article (what does this mean? ). If published, this will include your full peer review and any attached files.

**Do you want your identity to be public for this peer review?** For information about this choice, including consent withdrawal, please see our Privacy Policy .

Reviewer #2: **Yes: ** Marya Lieberman

Reviewer #3: **Yes: ** Thomas Bizimana

---

## [Editor Report · Decision Letter 2]

4 Feb 2025

A Comparative Study Between Near-Infrared (NIR) Spectrometer and High-Performance Liquid Chromatography (HPLC) on the Sensitivity and Specificity.

PONE-D-24-54221R2

Dear Dr. Maffioli,

We’re pleased to inform you that your manuscript has been judged scientifically suitable for publication and will be formally accepted for publication once it meets all outstanding technical requirements.

Kind regards,

Hope Onohuean, PhD

Academic Editor

PLOS ONE
---

## [Editor Report · Acceptance letter]

PONE-D-24-54221R2

PLOS ONE

Dear Dr. Maffioli,

I'm pleased to inform you that your manuscript has been deemed suitable for publication in PLOS ONE. Congratulations! Your manuscript is now being handed over to our production team.

Kind regards,

on behalf of

Dr. Hope Onohuean

Academic Editor

PLOS ONE